# A Review of Smart Lubricant-Infused Surfaces for Droplet Manipulation

**DOI:** 10.3390/nano11030801

**Published:** 2021-03-21

**Authors:** Zhentao Hao, Weihua Li

**Affiliations:** 1School of Chemical Engineering and Technology, Sun Yat-sen University, Zhuhai 519082, China; haozht@foxmail.com; 2Southern Marine Science and Engineering Guangdong Laboratory (Zhuhai), Sun Yat-sen University, Zhuhai 519082, China; 3School of Materials Science and Engineering, Sun Yat-sen University, Guangzhou 510275, China

**Keywords:** slippery surface, wettability, droplet manipulation

## Abstract

The nepenthes-inspired lubricant-infused surface (LIS) is emerging as a novel repellent surface with self-healing, self-cleaning, pressure stability and ultra-slippery properties. Recently, stimuli-responsive materials to construct a smart LIS have broadened the application of LIS for droplet manipulation, showing great promise in microfluidics. This review mainly focuses on the recent developments towards the droplet manipulation on LIS with different mechanisms induced by various external stimuli, including thermo, light, electric, magnetism, and mechanical force. First, the droplet condition on LIS, determined by the properties of the droplet, the lubricant and substrate, is illustrated. Droplet manipulation via altering the droplet regime realized by different mechanisms, such as varying slipperiness, electrostatic force and wettability, is discussed. Moreover, some applications on droplet manipulation employed in various filed, including microreactors, microfluidics, etc., are also presented. Finally, a summary of this work and possible future research directions for the transport of droplets on smart LIS are outlined to promote the development of this field.

## 1. Introduction

Since pioneering work by Aizenberg et al. in 2011 [1], the nepenthes-inspired lubricant-infused surface (LIS) is emerging as an excellent novel class of repellant surfaces containing a patterned hydrophobic substrate and nonreactive lubricant immobilized via van der Waals or capillary forces. Unlike a superhydrophobic surface, the interface between LIS, droplet and air is a relatively stable liquid-liquid-vapor rather than liquid-vapor-vapor. Thus, the LIS is atomically-smooth, self-healing properties, and pressure tolerance (~7 km of hydrostatic pressure) [1]. Besides, the chemically homogeneous lubricant with low surface energy and fluidity endows the LIS with defect-free, spontaneously smooth and repellant features, facilitating the realization of an extremely low contact angle hysteresis for most liquids. These unique features potentially address the drawback of the superhydrophobic surface (e.g., depletion of air cushion). Benefiting from these excellent properties, LISs are used to exploit different functions in a wide range of fields, such as anti-biofouling [2,3,4,5,6], corrosion resistance [7,8,9,10,11,12], ice/frost delaying [13,14,15,16,17], self-cleaning [18,19,20,21,22], water harvesting [23,24,25], drag reduction [26,27,28,29,30,31], heat transfer [32,33,34]. Some recent works give an overview of LIS’s overall development ranging from fundamental fabrication to various applications [35,36]. Recently, another application of LIS was developed by endowing the LIS with tunable wettability. This wettability-tunable smart LIS can dynamically control the mobility of droplets, demonstrating significant promising industrial applications (e.g., microfluidics, “no-loss” droplet transport). Accordingly, various methods have been deployed to achieve smart LIS by means of thermal, electrical, magnetic, photoinduced and mechanical actuation. Targeting this active direction, Yang et al. summarized developments of wettability-tunable LISs induced by external stimuli and their applications in fog collection, oil–water separation, complex-flow distribution, droplets’ delivery in 2019 [37]. However, with the fast advancement of smart LISs, many important and successive works on improvements of responsive LISs and other nonconventional LISs were developed for droplet manipulation [38,39,40,41,42,43,44,45,46,47,48,49,50,51,52,53,54,55], but these studies are not reported by Yang‘ work.

This review is an overview of the recent progress in smart LIS for droplet manipulation induced by external stimuli, as presented in Figure 1. We first give a summary of some unique properties and fundamentals of droplets on LIS. Then, the smart LIS for droplet manipulation is classified according to the external stimulus (e.g., thermal) and corresponding mechanisms (e.g., the difference of wettability). We highlight the promise of the smart LIS in droplet manipulation and compare the merits and deficiencies of the various smart LIS. Finally, we conclude this review by describing the challenges and presenting an outlook for future research in the smart LIS domain.

## 2. Fundamental of Droplet Mobility on LIS

### 2.1. Droplet Regime on LIS

The motion of a droplet is related to the dynamic friction factor (*μ*) and viscous force (*f*). The sliding or pinning of the droplet depends on its contact with LIS. A stable LIS should satisfy the following criteria [1]: (1) The lubricant, immiscible and nonreactive with external liquid, is stably immobilized by the substrate; (2) The affinity of the lubricant with the substrate is much firmer than with the external liquid. These criteria ensure that the stable presence of the lubricant layer is not replaced by the external fluid. Similar to hemi-wicking, the prerequisite for durable impregnation requires the contact angle of the lubricant to be smaller than a critical angle *θ_c_*, which can be described as follows:(1)$$\cos\theta_{c}=\frac{1-\varphi }{r-\varphi }$$
where *φ* is the fraction of the projected area of the roughness structure, and *r* is the roughness of the substrate.

For the region underneath the droplet, the submergence or exposure of protrusions can be predicted by the spreading coefficient of lubricant (*o*) on the rough substrate (*s*) in the presence of the droplet (*w*):(2)$$S_{os}\left(w\right)=\gamma_{sw}-\gamma_{os}-\gamma_{ow}$$
where *γ* is the interfacial energy.

When the surface energy of the substrate is high, the lubricant immobilized by the substrate is completely replaced by the droplet (W1 in Figure 2). This results in high friction between the droplet and the LIS [56]. With the surface energy of the substrate decreasing, the droplet only replaces the outer lubricant layer rather than the lubricant inside pores. (W2 in Figure 2). When the surface energy of the substrate is low enough, protrusions are covered with a lubricant layer, which cannot be replaced by the droplet (W3 in Figure 2). This can significantly decline the friction. Thus, the droplet is prone to sliding on an overcoated smooth lubricant layer at a low sliding angle.

### 2.2. Droplet Manipulation on LIS

Droplet manipulation could be classified in two cases: (1) the droplet is placed on a tilted surface, and (2) the unbalanced force applied on the droplet. In the first case, the droplet on a tilted surface with a tilting angle *α* deforms due to the friction. The contact angle hysteresis (Δ*θ* = *θ_a_* − *θ_r_*) originates from the inhomogeneous surface, including the roughness and surface energy, which is positively related to the friction. When *α* exceeds the critical sliding angle *α**, the droplet starts to slide. In this case of a droplet on tilted LIS, the driving force of the droplet moving on tilted slope arises from the gravity:(3)F=ρVgsinα
where *ρ*, *V*, *g,* and *α* represent the density, volume of the liquid droplet, gravitational acceleration, and the tilted angle, respectively. For a given tilted surface, the droplet motion’s driving force remains constant, as shown in Figure 3a. However, applying stimuli enables the control of droplet movement by changing the friction *(f)*. This friction (*f*) during sliding can be calculated as:(4)f=μN
where *μ* and *N* refer to the dynamic friction factor and normal force, respectively; *μ* is a positive relationship with the surface roughness.

Droplet motion on a flat LIS mainly requires an external stimulus to form the unbalance forces (e.g., Marangoni force, Young’s force), as shown in Figure 3b. Surfaces with high free surface energy are preferentially wet compared with low-surface-energy surfaces. As a result, the droplet on a LIS moved toward the high-surface-energy regions.

The Marangoni effect refers to forces induced by inhomogeneities in the free surface energy of the surface. External stimuli (e.g., electric fields, temperature gradients) can temporarily generate a gradient of surface energy defined as the Marangoni force (*F_M_*). The value of Marangoni force has relevance with the surface tension gradient, which can be expressed as follows [57]:(5)$$F_{M}\propto \pi R^{2}\left(\frac{d\gamma_{wa}}{dT}-\frac{d\gamma_{ow}}{dT}\right)\frac{dT}{dx}$$
where *R* is droplet radius, *T* refers to the driving force (e.g., temperature), the *dγ_wa_/dT* means the variation in droplet-air total interfacial tensions with the driving force, and *dF/dx* presents a gradient on the surface along the direction of droplet motion.

Young’s force occurs due to the asymmetric deformation of the droplet, and unbalanced Young’s force (*F_Y_*_._) acts on two sides of the droplet as a driving force to the droplet. *F_Y_*_._ is described as the following equation [40]:(6)$$F_{Y}=2R\left(\cos\theta_{r}-\cos\theta_{a}\right)\gamma_{wa}$$
where *γ_wa_* refers to the surface tension of the droplet in the air.

For droplets directly in contact with the lubricant layer, the sliding and pinning behavior is associated with the physical property of the lubricant or underlying substrate, which can be altered by an external stimulus.

## 3. Droplet Manipulation on Smart LIS

In general, approaches of droplet manipulation on smart LISs include varying slipperiness, electrostatic interactions, the difference of the wettability and other means, which have been reported in recent works. Details of these works are presented in Table 1 and they will be discussed in the following sections.

### 3.1. By Varying Slipperiness

Once droplets are placed on a LIS, they directly contact the lubricant layer. Therefore, one of the most straightforward and facile droplet manipulation methods is to tune the slipperiness of the lubricant by external thermal stimuli. Several suitable phase-change lubricants, such as paraffin, cocoa oil, tetracosane, and coconut oil, have moderate melting temperatures (35~60 °C) and become liquid under external heating. Thus, by heating the lubricants, the droplet is no longer on the solid-surface but on the liquid-surface, on which the droplet could slip and move.

Among the first generation of thermally responsive LIS for droplet manipulation was a classic example reported by Wang et al. [58]. By using n-paraffin as a lubricant and heating the n-paraffin, an n-paraffin-swollen organogel is obtained for switchable droplet motion manipulation. As shown in Figure 4a–c, above 58 °C (melting temperature), the droplet can gradually slide on the liquefied paraffin due to the reduced friction. In contrast, the droplet can be pinned below 58 °C. More recently, the same concept also expanded to the development of a thermal-responsive smart LIS for directional droplet motion manipulation [59]. Paraffin-infused directional porous polystyrene (PS) served as an anisotropic surface for directional droplet transportation. The grooved PS only allows the droplet to slide easily along the groove. These thermally responsive LISs show excellent reversibility to control water adhesion in cyclic tests.

However, one limitation of these LIS with the tunable wettability properties arises from a required ex situ heat source (e.g., hot plate, air oven). Thus, it is unrealistic to control a droplet’s behavior by repeatedly placing the sample near or away from the heat source. Moreover, an energy-consuming hot plate or oven regularly requires an input high voltage of 220 V. Integrating LIS with a conductive metal wire as an in situ electric heater could be a solution to in situ control of the droplet under a low voltage. The energized metal wire is capable of generating joule energy to heat and melt the paraffin layer. Zhao et al. constructed a hydrophobized porous SiO_2_ structure, in which a wire heater with the specific pathway was embedded, followed by paraffin infusion. Loading the voltage allows the melting of the surrounding paraffin, forming a fluid path along the wire. By tilting the LIS, the droplet can slide along the fluid pathway [41]. This technique was improved by integrating a flexible silver nanowire heater with the paraffin-infused micropillar-arrayed zinc oxide LIS [38]. By loading a 12 V voltage on the nanowires, the paraffin becomes melting, with the interface transition from an air–liquid–liquid to a lubricated air–liquid–solid one within 40 s.

Furthermore, the use of hybrid lubricant (a mixture of solid and liquid paraffin) can also achieve an ultra-low-voltage response and shorten response time [39]. Chen et al. demonstrated that a 5-V joule heating generated from a nanowire heater could liquefy above paraffin within 20 s forming a dynamic slippery surface, and afterward, the droplet starts to move (Figure 5a). Additionally, the voltage to melt the lubricant can be lowered to 2 V by tuning the liquid paraffin ratio. Because this hybrid paraffin has merits of good biocompatibility and low surface energy, this type of LIS not only can be used for intentional transport of nonconductive biological or conductive ionic droplets (NaCl) but also diverse organic droplets manipulation (e.g., glycerol, ethylene glycol, dimethyl sulfoxide), as shown in Figure 5b. Theoretically, the nanowire heater can be integrated with various patterned substrates, demonstrating adaption to various materials.

Light is another competitive stimulus to induce lubricant phase transition owing to its remote, excellent spatial and temporal control merits. A light-responsive slippery surface is typically realized by incorporating photothermal particles (e.g., Fe_3_O_4,_ carbon materials) into the substrate/lubricant during the fabrication process. Under near-infrared (NIR) light irradiation, the photothermal particles inside the substrate can absorb energy, generating heat and melt the lubricant. For instance, a LIS was designed by infusing low-melting cocoa oil (melting point: 35 °C) and efficient photothermal Fe_3_O_4_ nanoparticles into the anodized aluminum oxide substrate [44]. Under an infrared light beam irradiation, diverse droplets, such as blood, ink, milk, and ethanol, were successfully controlled to slide along the tilted LIS. Fe_3_O_4_ nanoparticles can absorb energy and melt the cocoa oil into slippery liquid oil. The radiated region’s temperature was shown to rise from −10 °C to 46 °C in ~70 s. Increasing the Fe_3_O_4_ contents can slightly enhance maximum achievable temperatures and lower the power of the light beam to achieve these temperatures. Using the photothermal graphene oxide (GO) as a raw material to construct a porous lamellar GO sponge also can be free from incorporating additional photothermal particles (Figure 6a–d) [43]. Similarly, melted paraffin at the irradiated region becomes slippery, whereas the unirradiated parts remain unchanged (Figure 6e–g). By masking some parts of the surface and leaving a specific uncovered pathway (e.g., straight line, arc, Y-shaped curve), it is possible to drive the droplet sliding along a predefined route (Figure 6h). A similar study used CNT incorporated into a matrix used as photothermal particles for droplet manipulation [42]. Considering easy volatilization of lubricant and the requirement of long-term lubricant service time, Sun et al. designed a high-boiling-point ionic phase-change lubricant, which was infused into a Fe_3_O_4_ nanoparticle-embedded hybrid self-repairable film [47]. It is found that Fe_3_O_4_ nanoparticles embedded in the hybrid film can absorb not only light but also energy from a magnetic field to melt the ionic lubricant. Thus, remote controlling droplet motion is realized in response to both responsive sources. The slippery surface can withstand 80 °C for 10 days without a change in wettability, showing exceptional stability to evaporation.

Like the light stimuli, magnetic stimuli are also attractive to achieve real-time wettability control because of the remote operability, rapid response, and high maneuverability. Compared with the thermal-induced controllable phase transition of lubricant, magnetic-stimuli LIS can instantly manipulate droplet motion without any delay. One of the facile ways to realize the magnetic droplet control is to embed the magnetic particles into the soft substrate of LIS. In the presence of a magnetic field, magnetic particles (e.g., Fe, Fe_3_O_4,_ cobalt) inside the soft substrate are attracted to form protrusions, which promotes surface friction, as shown in Figure 7a,b. The formation of the protrusions enables the transition of the surface from slippery to rough. As a commonly used elastomer, PDMS has become one of the best candidates to encapsulate the magnetic particles due to its inert, non-toxic, good biocompatibility properties. Guo et al. presented a magneto-controllable droplet on slippery gel surfaces, which were fabricated by integrating magnetically responsive Fe/PDMS gel films with silicone oil [46]. It is found that increasing the intensity of the magnetic field can lower the sliding angle and enhance hydrophobicity. Transportation and merging of the droplets for a microreactor were also demonstrated via a chromogenic reaction shown in Figure 7c. The yellow FeCl_3_ droplet gradually slides towards the colorless pyrocatechol on the tilted LIS. Once the two droplets start to contact, the magnetic field is loading for 9 s to ensure a sufficient reaction time, and the droplet turns dark. Removing the magnet allows the dark droplet to slide off the surface due to the inherent self-cleaning property of LIS. Notably, the slippery gel surface showed superior air stability after 100 cycles due to the molecule-level dispersion of silicone oil in the PDMS network [46].

Droplets can also be manipulated on the LIS by applying mechanical stimulation (e.g., stress tensile, acoustic wave) with the advantages of instant, facile, environment-friendly properties. External tensile stress reduces pressure inside the porous substrate, causing the depletion of the lubricant layer and enhanced friction, so the droplet is pinned on an oil-less rough surface. When releasing the LIS, the morphology returns to the nearly original state, the droplet continues to slide. Some of the elastomers, such as 1,2-polybutadiene, PDMS, propylene, polyurethane (PU), have been used as a flexible substrate to immobilize the lubricant layer. Aizenberg et al. used oil-infused porous Teflon to manipulate the droplet movement on a PDMS substrate [60]. Stretching or releasing the LIS caused the pinning or sliding of the droplet. Altering the volume fraction of lubricant on a flexible substrate can tune the droplet manipulation’s responsibility under deformation. Still, lubricant viscosity has little effect in terms of the response rate to the mechanical load. Moreover, stimulus sensitivity also can be manipulated by varying lubricants with different surface tensions.

In another example, 1,2-polybutadiene was replaced by a flexible PDMS substrate [45]. A moderate infused amount of lubricant is crucial for the adaptive performance of mechanical-responsive LIS. The ultra-low amount of the infused lubricant could not fill the pore in an elastic substrate film, so it fails to form a slippery surface. At the same time, excessive lubricant also compromises the ability of the LIS to tune wettability [60]. Wang et al. found that droplets on mechanical-responsive LIS can resist wind blowing during transportation. Wind blowing can serve as a mechanical stimulus, enabling the deformation of the flexible LIS and the withdrawal of the lubricant layer. This increases friction against wind [61]. Isotropic directional droplet transportation can be realized by creating a groove structure similar to other responsive LIS [51].

Considering the unrealistic fact that maintaining the wettability of flexible LIS requires sustained tensile forces, this field has gradually focused more on a facile external factor to induce the stretching/release of the flexible substrate to remotely control droplets. One such typical work was conducted by Hu et al., who designed an electrically induced stretchable LIS for droplet manipulation, inspired by muscles and tissues [49]. As illustrated in Figure 8a, a poroelastic silicone oil-infused polypropylene (PP) porous film is stuck on a dielectric elastomer sandwiched between two flexible electrodes for droplet manipulation. The electrostatic interaction between the oppositely charged conducting layers allows the deformation of the dielectric to stretch the poroelastic film, causing the droplet pinning. Unloading the voltage enables the PP porous film to restore its original shape, thereby enabling the droplet to slide (Figure 8b). Actuation strain acting on the PP porous film can be tuned by pre-stretching ratios. Conventional adaptive, responsive LIS often relies on various transitions at the atomic/molecular scale to enable macroscopic dynamic behavior. This muscle-mimicking LIS overcomes this constraint. However, to realize the horizontal expansion of the PP film, an ultra-high voltage (up to 12 kV) was applied to the two conductive flexible electrodes, which results in a high risk of electric shock. In contrast to the electrically-induced slippery film stretching, Zhao et al. demonstrated a slippery improvement by using a thermally expanded elastomer [50]. They presented a shape-memory graphene/trans-1,4-polyisoprene (TPI) hybrid sponge for dynamically manipulating droplets via temperature-induced expansion. After testing 20 cycles with strains up to 85%, this hybrid slippery sponge has shown durability and reversible compressive deformability. Because of the film’s electrothermal and shape-memory features, the presence of a constant DC voltage (6 V) applied to the graphene/TPI sponge enables the heat-induced vertical expansion of the sponge. After unloading the voltage, the nature of the shape memory property allows the restoration of the hybrid film to a slippery state.

### 3.2. By Electrostatic Force

Utilizing electrostatic interactions between the substrate and the droplets also allows the control of droplet sliding and pinning behavior. Such type of electric-stimuli LIS requires a conductive droplet and conductive substrate working as the medium, a dielectric layer (lubricant layer) as a capacitance between the droplet and substrate [48]. Metal and alloys, carbon materials (e.g., graphene oxide, carbon nanotubes), and conductive polymers (e.g., polypyrrole) have been used for conductive substrates. Still, substrates with low conductivity (semiconductor and isolator) are also applicable after modification by conductive substances, such as Ag or indium tin oxide (ITO). As shown in Figure 9a, after applying a voltage, the conductive substrate and droplet generate opposite charges, enabling the electrostatic force to enhance the friction. Heng et al. explored the droplet motion on tilted conductive polymer-based LIS [62]. They found that by applying a voltage, the droplet is pinned, and removing the voltage allows the droplet to slide, driven by gravity (Figure 9b–d). Physical properties (e.g., conductivity, viscosity, thickness) of the lubricant layer and the type of droplet also play an essential role in the electrostatic force between the lubricant and droplets [63,64]. Using a high conductivity lubricant (e.g., ionic liquid) can generate a high charge density and an enhanced interaction compared to a nonconductive lubricant (e.g., silicone oil).

Moreover, increased viscosity of the lubricant results in a weakened electrostatic interaction due to a reduction of the permittivity. Enhanced applied voltages are needed to achieve the pinning of the droplet. The friction between the droplet and the LIS increases with the applied voltage. However, for nonconductive droplets (e.g., glycerol, ethylene glycol), applying a voltage results in little effect on the droplet sliding control because these organic droplets have high resistance. Regardless of the type of infused lubricant, nonconductive droplets cannot be controlled on LIS. Heng et al. further used a photoelectric substrate to construct a LIS, achieving a photoelectric responsive approach of droplet manipulation [65]. Increasing the intensity of UV light can enhance electrostatic force to pin the droplet on a tilted slippery surface. However, the light beam radiation is only limited to high-energy UV for achieving adequate electrostatic force.

### 3.3. By the Difference of the Wettability

Though the thermal-induce lubricant phase transition and electrostatic force have achieved success in droplet manipulation, still, the driven force by gravity is limited in some cases. To enable the droplet sliding, the sample should be tilted to a certain angle in advance so that the droplet can start to slide. Some solutions have been exploited to address this deficiency.

One solution is to create a temperature gradient by taking advantage of the synergistic effect of Young’s force (*F_Y_*) and Marangoni force (*F_M_*). When doping photothermal particles into the LIS, the temperature rises in the irradiated region and the surface tension decreases. A localized dynamic temperature gradient causes the surface tension difference and the Marangoni force between the irradiated side and the other side of the droplet, which induces the droplet movement towards the unirradiated side (Figure 10a). For example, a photothermal-responsive organogel surface was developed for droplet control based on the temperature gradient. This organogel-based LIS is achieved by embedding Fe_3_O_4_ nanoparticles in PDMS, followed by silicone oil infusion. Figure 10b shows the local change in temperature at the droplet when irradiated. The synergistic effect of *F_Y_* and *F_M_* provides a powerful driving force so that a 7 µL water droplet can move on the LIS tilted at a 10° angle against the slope. A droplet can be moved towards arbitrary directions in the presence of a unilateral NIR stimulus (Figure 11a). The droplet marked as No.1 was manipulated to slide and sequentially merged with other droplets. Moreover, the conductive NaCl droplet was also driven to turn on the disconnected circuit, enabling the diode to emit a red light (Figure 11b) [40]. More recently, Wu et al. further investigated the impact of lubricant viscosity, surface tension, and Fe_3_O_4_ content on *F_Y_* and sliding velocity. Young’s force (*F_Y_*) is enhanced with the increase of the droplet’s surface tension and amount of doped Fe_3_O_4_. However, increasing the lubricant viscosity lowers the *F_Y_*, leading to a decreased sliding velocity [54]. To effectively generate the photothermal effect, the use of long-wavelength near-infrared light with strong penetration is necessary to irradiate the particles inside the substrate. The limitation lies in that only a few photothermal particles, such as graphene and Fe_3_O_4,_ can respond to near-infrared light irradiation.

Another feasible solution is to precisely design a conductive array substrate, where the insulating and conducting regions appear alternately. As presented in Figure 12, the LIS is used as an electrode in an electrowetting-on-dielectric (EWOD) microfluidic device to encapsulate the droplet. Both sides of the droplet are located in the conductive region. The electrostatic force at the energized conductive area is more extensive than the unenergized one, making an easier decline of contact angle at the energized conductive region. The relationship can be described with the following Young–Lippmann equation [48]:(7)$$\cos\theta_{v}=\cos\theta_{0}+ф\cdot \frac{\varepsilon \varepsilon_{0}}{d}\frac{U^{2}}{\gamma_{wo}}$$
where *θ*_0_ (°), *θ_v_* (°) correspond to the contact angle before and after applying the voltage *U* (V), *γ* (N⋅m^−1^) is the interfacial tension between the conductive droplet and surrounding medium, *ε*, *ε*_0_ and *d* (m) refer to the dielectric constant, the permittivity of vacuum (8.85 × 10^−12^ F⋅m^−1^), and the thickness of a dielectric layer, respectively. Thus, the resulting force difference between the two sides of the droplet can induce the droplet movement in the direction towards which the contact angle decreases [48]. Cao et al. used this method to move a droplet of NaOH towards another phenolphthalein droplet (Figure 12e). With the assistance of applied voltage (350 V), a 1.0 μL drop of NaOH aqueous solution with the concentration of 0.001, 0.01, 0.1, 1 M, was gradually actuated to merge with phenolphthalein (0.5 wt %) and finally, the merged large droplet turned pink. Figure 12d shows a red droplet (1.0 μL) manipulation on both convex and concave surfaces. This indicates this surface can potentially serve as a highly flexible and reliable microfluidic device for droplet manipulation. Electric-stimuli-responsive smart LIS is a promising candidate to control droplet motion. It is worth mentioning that to pin the droplet requires a certain high voltage. However, applying excessive voltage can break through the insulating lubricant layer, which results in the failure of droplet control. Additionally, both the droplet and substrate need to be conductive, limiting the scope of application of this technique.

Liu et al. found when a droplet is injected on the surface, and it spontaneously moves towards the opposite direction of the tilted microcilia (Figure 13a) [68]. The water sliding direction can be instantly guided by adjusting the direction of the microcilia tilt. The direction of the microcilia tilt can be controlled by magnetic field direction. Based on this phenomenon, a device for droplet transportation was designed, as shown in Figure 13b. A water droplet located on the hydrophilic–hydrophobic boundary can be directionally transported and continuously guided into the hydrophilic channel. Different colored water flows can be separated and directionally transported to the desired channel by altering the magnetic field direction, demonstrating a real time, controllable, and unidirectional water delivery in a specific direction (Figure 13c). The driving force to motivate water droplets in this work relies on LIS’s difference and filter paper’s wettability, so an external hydrophilic surface is needed to assist the droplet transportation.

Applying a gradient magnetic field to form a wettability difference can also achieve droplet transportation in a fast, convenient manner. The mechanism is presented in Figure 14 [66]. The intensity of the magnetic field can tune the interface structure toward hydrophobicity due to the reorganization of the magnetic particles inside the lubricant layer (Figure 14a). In agreement with the study by Heng et al. mentioned above [46], the hydrophobicity increases with the intensity of the magnetic field. Therefore, the presence of a magnetic field gradient results in a difference of the wettability at the two sides of the droplet and serves as a driving force for the movement towards the direction of the small contact angle, as shown in Figure 14b,d. The adaptive surface can restore its original state (Figure 14c).

### 3.4. By Other Means

Guo et al. presented another LIS, composed of a liquid-repelling surface, a magnetic actuating layer, and a PDMS substrate. It allows for omni-liquid droplet manipulation without tilting the LIS, as shown in Figure 15a [52]. When a magnetic probe is near the LIS bottom, magnetic attraction creates a small depression towards, which the droplet is driven by the slope. Thus, the sliding route of droplet motion can be guided by a magnetic probe in an arbitrary direction.

In addition to reconstructing the surface structure, another possible manner to realize droplet manipulation magnetically is by endowing the droplet with magnetism. By adding magnetic particles (MP) into the droplet, droplets can be driven to move and be guided by a magnet. Agrawal et al. used this method to achieve an On-chip synthesis of the Nylon-66 [55]. First, all the droplets become magnetic when Fe_3_O_4_ particles were added. Then, a droplet dissolved with hexamethyl diamine is driven to merge with a 10 μL NaOH aqueous droplet by a magnet. Finally, a hexane droplet dissolved with diacid chloride is magnetically actuated to mix with the merged hexamethyl diamine/NaOH droplet at ambient temperature. However, in this method, low MP concentrations and high driver magnet speeds result in the disengagement of MPs from the droplet. MPs are easily extracted from low-surface-tension droplets. Moreover, increasing the droplet volume generally reduces the droplet moving speed and MP concentration.

The droplet motion control also can be enabled by the mean of surface-acoustic-wave (SAW). Guo et al. presented a surface-acoustic-wave device for droplet motion based on silicone oil-infused porous hydrophobic silica integrated with interdigital transducers (IDTs) on a piezoelectric ZnO/Al platform [53]. After turning on the power of SAW, some of the energy and momentum from the longitudinal wave was dissipated into the droplet. The acoustic wave energy and momentum were transferred to the droplet, which forms a pressure or a body force to move the droplet. The threshold power to move the droplet decreases with increasing the silica layer thickness, while droplet motion on LIS with increasingly thick oil layers requires more energy. After 100 times cycle test, the threshold power for droplet transportation remains nearly unchanged, exhibiting a long-term performance. Droplets actuation induced by SAW demonstrates significant advantages of easy processing, tunable frequency response, and precise control. However, the deficiency is the easy detachment of the LIS from the piezoelectric ZnO/Al platform.

Wang et al. found chain configuration of the single-stranded DNA (ssDNA) can reversibly deform in response to the temperature [67]. This chain deformation has an impact on the exposure of hydrophobic moieties on the ssDNA chain, which strongly affects the interfacial hydrophobic interaction with the lubricant. Based on this property, ssDNA droplet motion can be controlled by temperature increase or decrease. For example, when the temperature increases from 283 to 303 K, the hydrophobic interaction between hydrophobic moieties inside the droplet and n-dodecane (lubricant) is suppressed (Figure 15b). This reduces interfacial adhesion between the droplet and the LIS. Therefore, the ssDNA droplet slides on the slippery surface. Compared with other thermal-responsive LISs, the limitation lies in only the biological molecules applicable rather than diverse droplets.

## 4. Conclusions

This work reviews the recent advances in droplet manipulation on stimuli-responsive smart LIS. From the perspective of a droplet property on LIS, substrate roughness and lubricant physical property have a major impact on the droplet motion, which is crucial for droplet manipulation. According to the droplet control mechanisms, approaches include slipperiness control, wettability control, and attraction force actuation. Smart LISs have succeeded in manipulating droplets, while some challenges still remain. Concerning the thermal or light-stimuli LIS, the long-term droplet manipulation is likely to suffer from the evaporation of the lubricant layer and the microdroplet due to the heat source. Electric-responsive LIS can perform droplet manipulation at ambient temperature, but a nonconductive organic droplet cannot be used, and a high voltage is necessary. As for mechanical stimuli-responsive LIS, to ensure droplet manipulation, the elastomer cracking during the operation should be considered. Hence, it is a worthwhile devoting effort to develop elastomer with high tensile strength, self-healing, and fatigue-resistant properties. Magnetic-responsive LIS may be promising surfaces because of the diverse and flexible manners they provide to manipulate droplets. Moreover, nearly all LISs face the challenge of depleting the lubricant. During the droplet movement, the loss of the lubricant layer caused by shear force is unavoidable for most lubricants, leading to a decrease in slipperiness and thereby an increase in friction. Moreover, droplets on LIS are likely to be contaminated by the underlying lubricant layer due to the low density of the lubricant and low surface tension. In microreactor and microfluidics, ensuring the high purity of the droplet during the transportation is required. Hence, creating a new lubricant class with a high boiling point and high density is of interest to perform the long-term droplet manipulation with slippery surfaces. Ionic lubricants as a stable layer in LIS have received attention as they are nonvolatile, non-flammable, exceptionally stable and have high boiling points. Ionic liquids are composed of anionic (e.g., AlCl^−^_4_, ZnCl_3_^−^, BF_3_^−^, PF_6_^−^) and cationic type (e.g., alkyl quaternary ammonium, alkyl quaternary phosphine, alkyl imidazole, alkyl pyridine) ion, which is bound by a strong Coulomb force, endowing the ionic lubricant excellent thermal stability. For example, Ma et al. synthesized a quaternary ammonium-based ionic lubricant, the decomposed temperature, which can achieve ~270 °C [69]. Lubricants containing ionic liquid show much lower friction.

In summary, although challenges in stimuli-responsive LIS still remain, we believe that their easy operation, as well as multiple stimuli sources in terms of various conditions, brings a bright future for droplet manipulation and related applications. We hope this review could provide a basis and guidance in feasible design and application at the industrial level.

## Figures and Tables

**Figure 1 nanomaterials-11-00801-f001:**
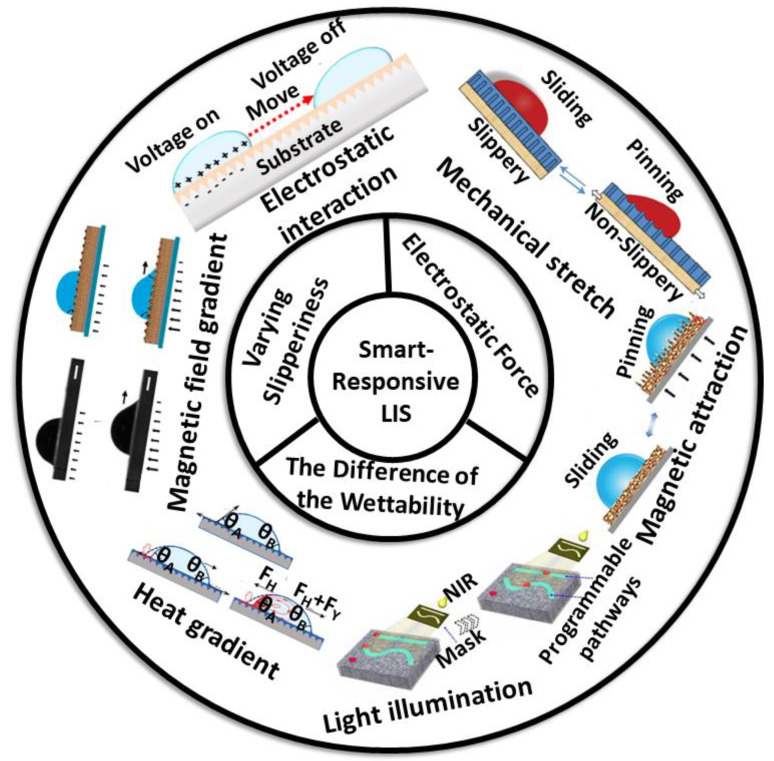
Overview of the droplet manipulation on a smart lubricant-infused surface (LIS).

**Figure 2 nanomaterials-11-00801-f002:**
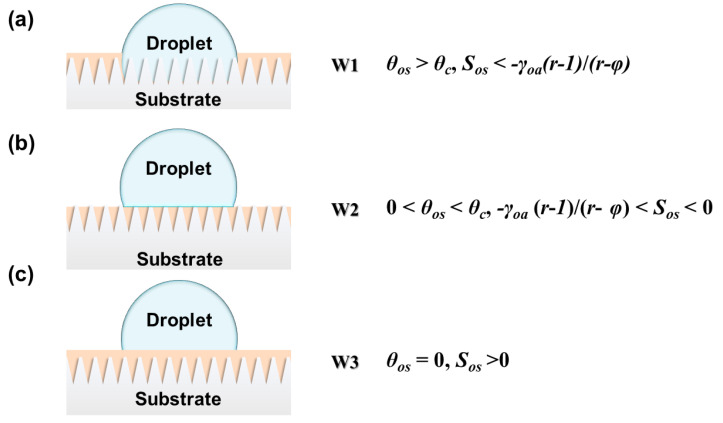
Scheme of (**a**) a droplet replacing the lubricant inside the roughness wet by the substrate; (**b**) wets the solid with a nonzero contact angle; (**c**) the droplet fails to replace the lubricant.

**Figure 3 nanomaterials-11-00801-f003:**
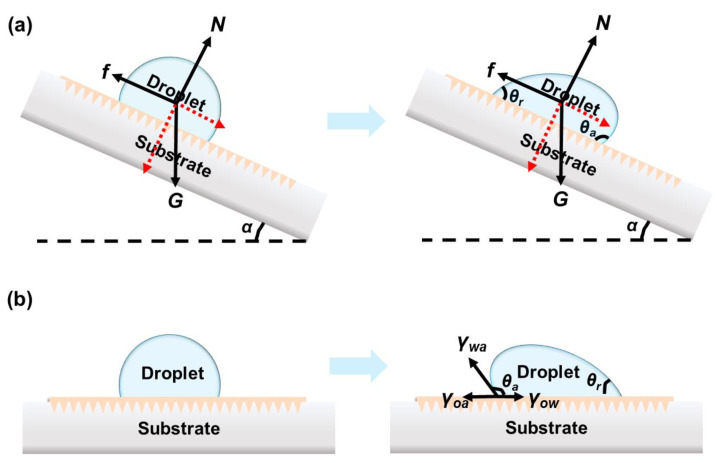
(**a**) Force analysis of the droplet on tilted LIS. *N, f, G* and *α* refer to the normal force, resistance induced by contact angle hysteresis, gravity and tilted angle, respectively. (**b**) Scheme of the driving force generated by the surface tension difference on the droplet to slide at flat LIS.

**Figure 4 nanomaterials-11-00801-f004:**
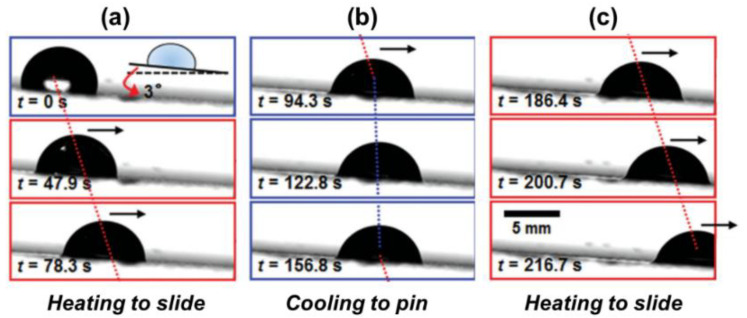
Demonstration of thermo-controllable drop sliding motion. (**a**) The LIS was heated to motivate the droplet sliding. (**b**) The droplet was pinned after removing the heat source. (**c**) The droplet continued to slide. Reprinted with permission from [58]. Copyright (2014) Wiley-VCH.

**Figure 5 nanomaterials-11-00801-f005:**
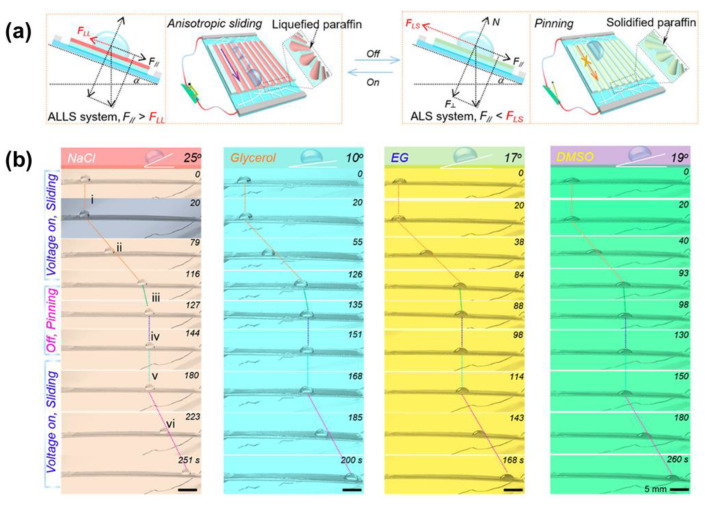
(**a**) Schematic of the mechanism to manipulate droplets thermally. The energized nanowire produces the joule heat to melt the paraffin and forms a liquified slippery paraffin layer with low friction. In this case, F > f, F and f refers to the driving force along the slope induced by gravity and friction force, respectively. This enables the droplet sliding. When the voltage is off, the liquid paraffin solidifies, so the friction increases, leading to the droplet pinning. (**b**) Demonstration of various droplets (NaCl, glycerol) manipulation, including NaCl, glycerol, ethylene glycol, dimethyl sulfoxide. Reprinted with permission from [39]. Copyright (2020) American Chemical Society.

**Figure 6 nanomaterials-11-00801-f006:**
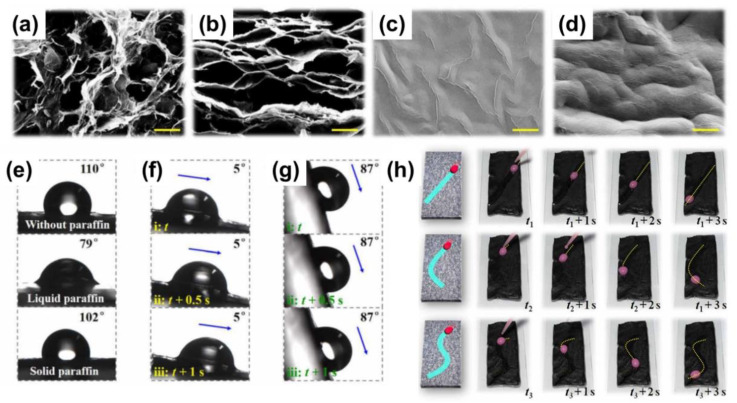
SEM (**a**) and cross-section (**b**) of a graphene sponge film; SEM (**c**) and cross-section (**d**) of a paraffin-infused graphene film. (**e**) Water contact angles of the porous graphene sponge film, the paraffin-infused graphene film in the presence and absence of radiation, respectively. (**f**) Demonstration of the droplet sliding on the paraffin-infused graphene film under the radiation of laser. (**g**) Demonstration of the droplet sliding on the paraffin-infused porous graphene film in the absence of the radiation. (**h**) Demonstration of the droplet sliding on a predefined route. The scale bar is 10 μm. From [43]. Reprinted with permission from AAAS.

**Figure 7 nanomaterials-11-00801-f007:**
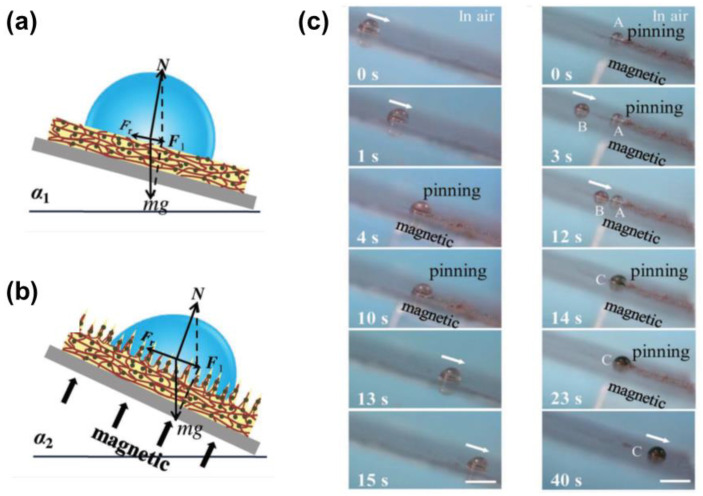
Scheme of a droplet’s force analysis before (**a**) and after (**b**) applying a magnetic field. N, F_r_, g, m, and F_1_ is the positive direction force refers to the normal force, surface retention force resulting from contact angle hysteresis (CAH), the mass of the droplet, the gravitational acceleration constant, and the resultant force of mg and N, respectively. (**c**) Droplet motion control on slippery gel surface and demonstration of the magnetically controlled liquid droplet transport for a chemical reaction. Reproduced with permission [46]. Copyright 2019, Wiley-VCH.

**Figure 8 nanomaterials-11-00801-f008:**
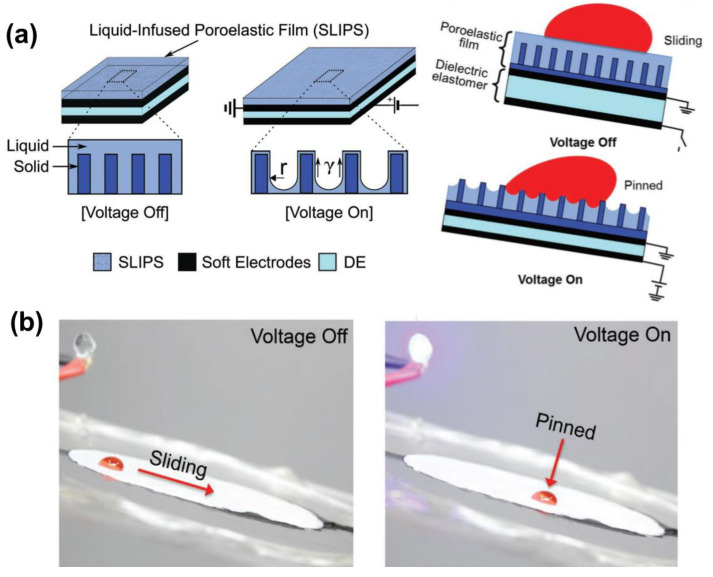
(**a**) Schematic illustration of electric-stimuli LIS and the droplet control mechanism. This smart LIS is integrated with the soft electrodes-sandwiched dielectric elastomer. The droplet initially slides on the tilted LIS in the presence of the lubricant layer when voltage is off. However, after applying a voltage, the poroelastic polypropylene is electrically stretched to reduce the pressure inside the pores, so the lubricant layer disappears and thereby, the droplet directly contacts the protrusions which leads to the droplet pinning. (**b**) Demonstration of the control of the mobility of a 50 µL droplet on a dynamic silicone-oil-infused polypropylene porous film tilted at 7° by loading/unloading 12 kV voltage. Reproduced with permission [49]. Copyright 2018, Wiley-VCH.

**Figure 9 nanomaterials-11-00801-f009:**
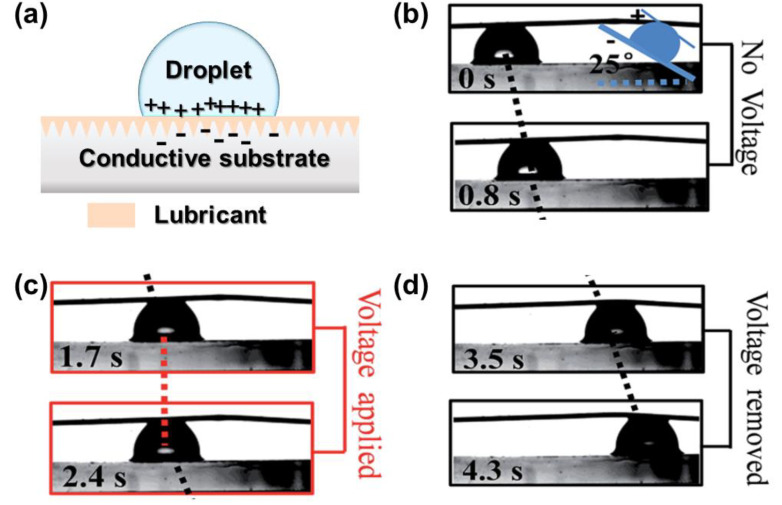
(**a**) Demonstration of the droplet control mechanism on electric-responsive LIS positive charges and negative charges are distributed in conductive droplet and substrate, respectively. (**b**) Water droplet sliding on LIS with oil viscosity of 100 cSt. (**c**) Droplet pinning on LIS when applying 2.4 V. (**d**) Droplet continues to slide after removing the voltage. The LIS sample is tilted at 25°. Reproduced from [62] with permission from The Royal Society of Chemistry.

**Figure 10 nanomaterials-11-00801-f010:**
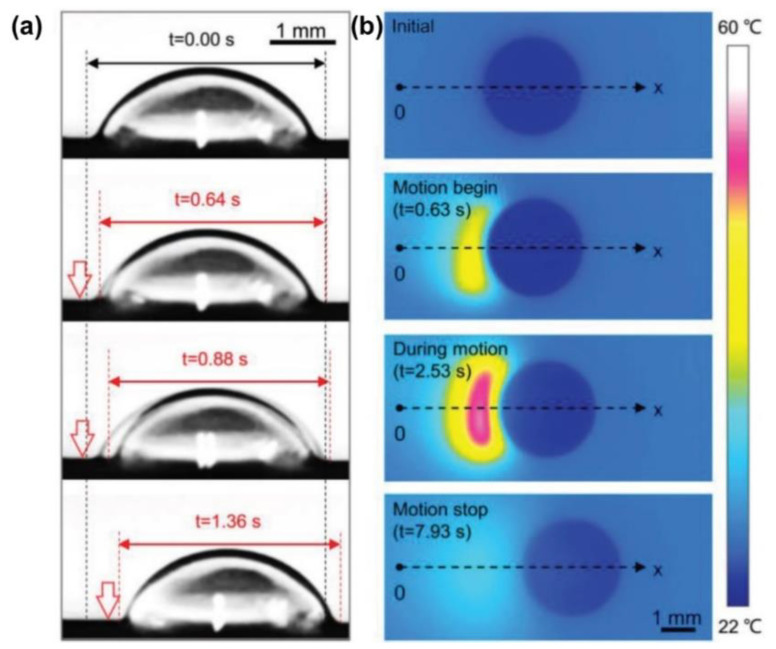
(**a**) Images of a propylene glycol droplet moving. The droplet’s left edge begins to shrink and moves towards the right under NIR. (vertical arrows). (**b**) Infrared images of 9 μL water droplet on LIS The irradiated droplet begins to move along the x-axis at t = 0.63 s and stops at t = 7.93 s after NIR irradiation is switched off. Reprinted with permission from [40]. Copyright (2018) American Chemical Society.

**Figure 11 nanomaterials-11-00801-f011:**
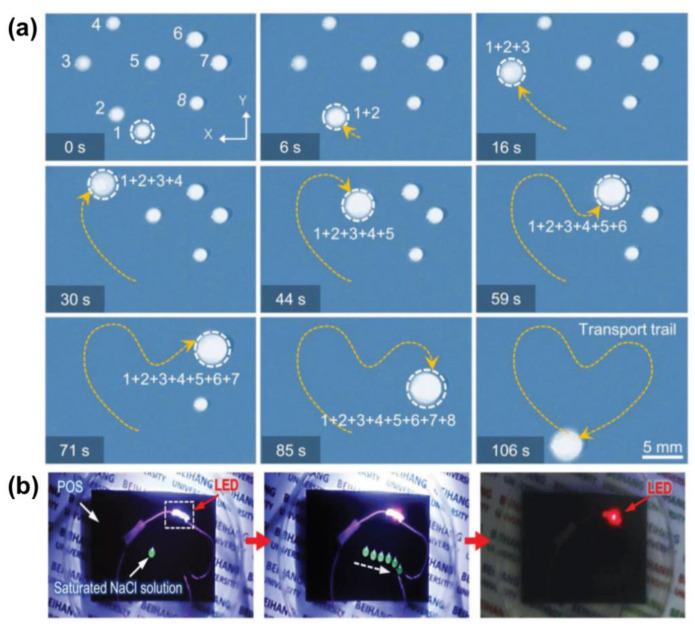
(**a**) Sequence of images of light-driven droplets on LIS There are eight water droplets (numbered with 1–8, the volume of each droplet varies between ≈3 and 5 μL). Droplet 1 can be moved to merge sequentially with other droplets along a designed “heart”-shaped trajectory in 106 s and finally form a large droplet. (**b**) Smart light-emitting circuit of droplet transport by NIR. The open-circuit is connected once a NIR guided droplet (≈6 μL) of saturated sodium chloride solution touches both ends of the circuit. The light-emitting diode in the circuit is turned on and gives off red light. Reprinted with permission from [40]. Copyright (2018) American Chemical Society.

**Figure 12 nanomaterials-11-00801-f012:**
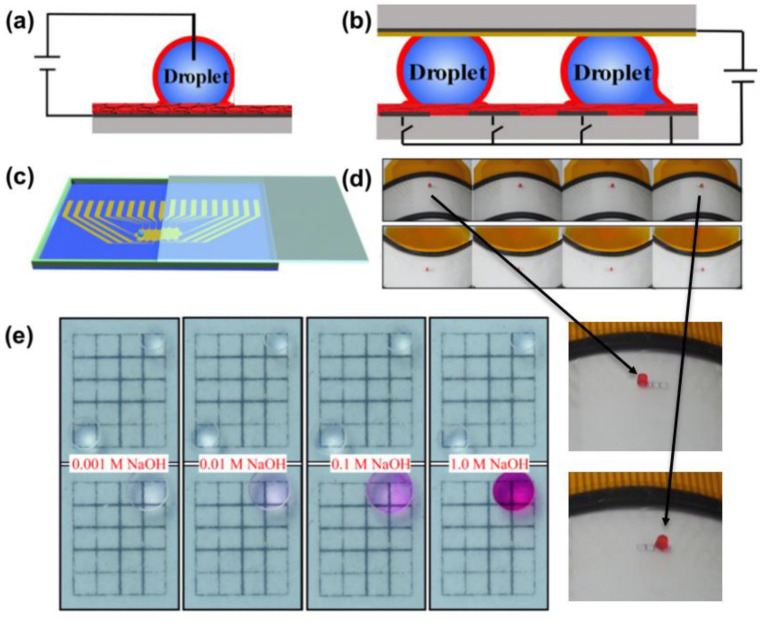
(**a**) Side-view image of a droplet on a slippery surface. The droplet was encapsulated by lubricant (red color). (**b**) Schematic of the electrowetting-on-dielectric (EWOD) microfluidic device. The insulative and conductive region is alternatively distributed. (**c**) The EWOD microfluidic device was with a droplet sandwiched by the top and bottom plates, driven by voltages applied to the patterned electrodes. (**d**) Demonstration of driving a 1.0 μL droplet to slowly move between 4 electrodes on convex and concave surfaces, respectively and corresponding enlarged views. (**e**) Side view image of a droplet on EWOD Images of actuating 1.0 μL NaOH aqueous droplet (0.001, 0.01, 0.1, and 1.0 M, respectively) at the bottom left corner to move and mix with a 1.0 μL phenolphthalein droplet (0.5 wt %). Reprinted with permission from [48]. Copyright (2019) Elsevier B.V.

**Figure 13 nanomaterials-11-00801-f013:**
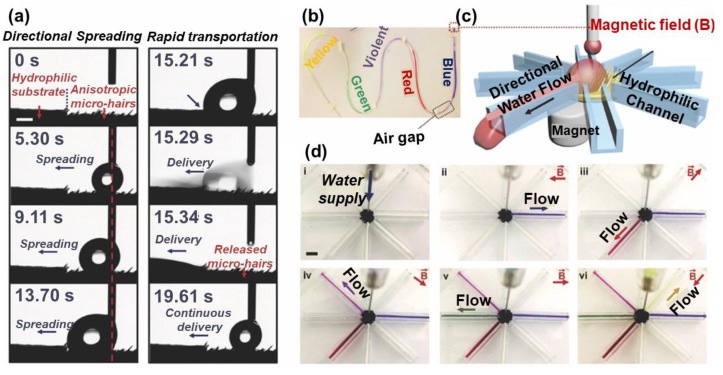
(**a**) Images of droplets moving towards a hydrophilic filter, away from the tilted microcilia. (**b**) Scheme of a water flow distributor, consisting of an inspirator underlying the LIS and 8 hydrophilic channels. (**c**) Illustration of controlling droplet sliding direction Water dyed with different colors was delivered through a plastic pipe and contacted the surface’s ferromagnetic microcilia surface. By varying the direction of the magnetic field, the tilting direction of the microcilia is changed, which switches the water flow in a defined direction. Reproduced with permission [68]. Copyright 2017, Wiley-VCH.

**Figure 14 nanomaterials-11-00801-f014:**
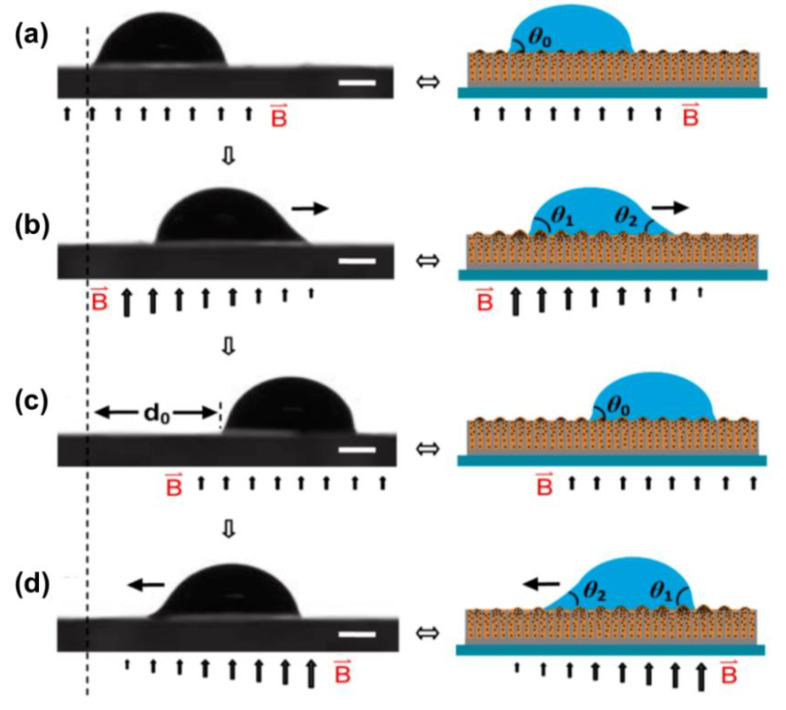
(**a**) A water droplet on the horizontal slippery surface under a uniform magnetic field. (**b**) The roughness gradient on the substrate surface induced by a gradient of the magnetic field results in the asymmetric deformation of the droplet. An unbalanced force caused by the roughness gradient drives the droplet to move. (**c**) The gradient magnetic intensity is transformed to a uniform one, so the droplet is pinned. (**d**) Applying an opposite-direction gradient magnetic field allows the droplet to move back. The water droplet size is ∼5 μL. The bars in (**a**–**d**) are 500 μm. Reprinted with permission from [66]. Copyright (2016) American Chemical Society.

**Figure 15 nanomaterials-11-00801-f015:**
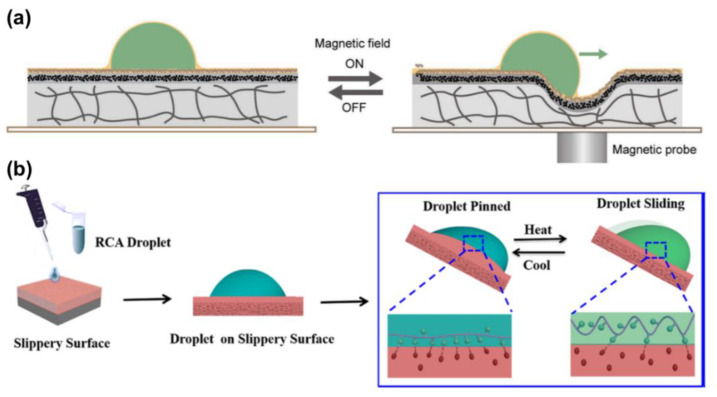
(**a**) Mechanism of the slippery surface to drive droplets by a magnet probe. Reproduced with permission [52]. Copyright 2019, Wiley-VCH. (**b**) Thermally control RCA biological droplet sliding behavior on the LIS. Reprinted with permission from [67]. Copyright (2019) American Chemical Society.

**Table 1 nanomaterials-11-00801-t001:** Overview of recent advances in droplet manipulation on smart LIS.

Authors	Lubricant	Mechanism	Response	Control Manner	Advantages	Limitations	Ref
Yao et al.	Paraffin	Phase transition of the lubricant	Thermal	Contact	Facile switchable process between low and high adhesion	Ex situ heat source required; not suitable for volatile droplets; non-instant response	[58]
Wang et al.	Paraffin	Phase transition of the lubricant	Thermal	Contact	Facile switchable process for directional control droplet	Ex situ heat source required; high-energy consumed; non-instant response	[59]
Chen et al.	Paraffin	Phase transition of the lubricant	Electrically thermal	Contactless	Directional control diverse droplet by in situ heating; remote droplet control	Not suitable for volatile droplets; non-instant response.	[39]
Gao et al.	Paraffin	Phase transition of the lubricant	Electrically thermal	Contactless	Directional diverse droplet control by in situ heating; remote droplet control	Easy volatilization of droplets; non-instant response.	[41]
Chen et al.	Paraffin	Phase transition of the lubricant	Electrically thermal	Contactless	Directional control droplet by in situ heating; remote droplet control	Easy volatilization of droplets; non-instant response.	[38]
Wang et al.	Paraffin	Phase transition of the lubricant	Photothermal (NIR)	Remote contactless	Noncontact regulation; spatial and temporal droplet control	Easy volatilization of droplets; non-instant response.	[43]
Wu et al.	Cocoa oil	Phase transition of the lubricant	Photothermal (IR)	Remote contactless	Noncontact regulation; spatial and temporal droplet control	Easy volatilization of droplets; non-instant response.	[44]
Li et al.	Paraffin	Phase transition of the lubricant	Photothermal (NIR)	Remote contactless	Directional droplet manipulation according to the patterned pathway.	Easy volatilization of droplets; non-instant response.	[42]
Rao et al.	Fluorinated ionic liquids	Phase transition of the lubricant	Photothermal (Sunlamp) and magnetic-thermal	Remote contactless	Dual responsive source applicable in complex environments	Small range of material selection; complex fabrication	[47]
Guo et al.	Silicone oil	Switchable regulation of protrusions	Magnetic	Contactless	Long operation time; excellent stability in air and underwater;	Limited well-organized microstructure; easy destruction of the soft substrate.	[46]
Kamei et al.	Fluorinated lubricant	Switchable regulation of the lubricant layer	Mechanical	Contact	Tunable, programmed repellency; spatiotemporal on-demand droplet manipulation	Poor strength and aging performance of substrates; requirement of sustained forces	[45]
Liu et al.	Krytox 103	Switchable regulation of the lubricant layer	Mechanical	Contact	Fast response; self-reporting; real-time monitoring wettability	Low robust-ness; requirement of sustained forces	[60]
Wang et al.	perfluoropolyether	Switchable regulation of the lubricant layer	Mechanical (wind)	Contactless	Fast response; wind blowing resistance of droplet	Not resistant to strong polar, acid and alkali droplets	[61]
Zhang et al.	Silicone oil	Switchable regulation of the lubricant layer	Mechanical	Contact	Fast response; facile fabrication; isotropic and anisotropic directional droplet manipulation	Low robustness; requirement of sustained forces	[51]
Oh et al.	Silicone oil	Switchable regulation of the lubricant layer	Electro-mechanical	Contactless	Multiple droplet behaviors manipulation (pinning, free sliding, repetitive stick–slip motions, extremely fast sliding, droplet oscillation, jetting, and mixing,	Extreme high voltage input	[49]
Wang et al.	Perfluorooctyl trichlorosilane	Switchable regulation of the lubricant layer	thermomechanical	Contactless	Flexible operation of stretching the films; low voltage input	Small range of material selection	[50]
Wang et al.	Silicone oil	Electrostatic attraction	Electric	Contactless	Instant response; tunable, programmed, on-demand droplet manipulation	A high dielectric constant is required. Only conductive substrate and droplet applicable	[62]
Cao et al.	Silicone oil	Wettability control	Electric	Contactless	Strong driving force; fast response	Only conductive substrate and droplet applicable	[48]
Che et al.	Silicone oil	Electrostatic attraction	Electric	Contactless	Tunable, programmed, on-demand directional droplet manipulation	Only conductive substrate and droplet applicable	[63]
Guo et al.	Silicone oil	Electrostatic attraction	Electric	Contactless	Directional droplet manipulation; instant response	Only conductive substrate and droplet applicable; harsh fabrication process;	[64]
Wang et al.	Silicone oil	Electrostatic attraction	Photoelectric	Remote contactless	Dual responsive remote droplet control	Only conductive substrate and droplet applicable	[65]
Wu et al.	Silicone oil	Wettability control	Photothermal	Remote contactless	Facile approach and droplet manipulation in arbitrary directions	Not applicable for high-viscosity droplets	[54]
Tian et al.	Silicone oil	Wettability control	Magnetic	Contactless	Fast response rate; controllable transport speed and direction	Fragile characteristics and easy damage of surface patterns;	[66]
Guo et al.	Fluorinated oils	Fast concaving under a magnetic field to transport the droplet.	Magnetic	Contactless	Suitable for sensitive droplets. rapid, reversible, and precise all types of droplets manipulation	Poor practical operability: a magnet probe required for indirect droplet manipulation;	[52]
Wang et al.	n-dodecane	Molecular configuration reversible deformation	Thermal	Contact	Facile process; even a general LIS applicable	Only a small part of biological droplet with a small application range	[67]
Luo et al.	Silicone oil	Momentum transfer	Mechanical	Contact	Tunable frequency response; precise droplet manipulation	Instability of the integrated LIS liquids with a low surface tension	[53]

## Data Availability

Data is contained within the article.

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
