# Peer review of "A Review of Smart Lubricant-Infused Surfaces for Droplet Manipulation"

_nanomaterials, 2021, doi:10.3390/nano11030801_

Round 1
Reviewer 1 Report
This paper is a small review of the recent publications related to the active manipulation of liquid droplets on lubricant infused surfaces. It is not the first paper on this subject, for example Yang et al. (https://doi.org/10.3389/fchem.2019.00826) published a paper on a similar subject in 2019. However, there is clearly a strong interest currently for droplet microfluidics and many papers have been published in the last two years on lubricant infused surfaces.
The paper is interesting and sums up the recent research in the field. The only problem is it is extremely difficult to read and significant effort needs to be made by the authors to improve it before publication. Many paragraphs should be totally re-written as they are impossible to understand. Spell-checking should be done before submission and the authors should ask someone, possibly someone whose native language is english and who is used to scientific writing, to read and correct their text without complacency before submission.
I made a few annotations on the manuscript submitted by the authors, which I join to this review, hoping it will help them implement corrections.

Reviewer 2 Report
The review by Hao and Li comprehensively summarises the progress in lubricant-infused surface (LIS) for droplet manipulation. After a brief introduction on the fundamentals of droplet manipulation, the review systematically discusses LIS based on the mechanism for droplet manipulation. Overall, this is an interesting and well-structured review paper which could stimulate the propagation of new concepts and approaches for LIS-based droplet manipulation.
There are some points which should be considered by the authors:
1) I believe this review would benefit from a table which could compare and provide an overview of the discussed droplet manipulation approaches. The table should include the advantages and limitations of each approach. Also, some indications about robustness and reliability of these approaches is an added value for readers.
2) Similar reviews have been published recently (e.g., Li et al., Slippery Lubricant‐Infused Surfaces: Properties and Emerging Applications, https://doi.org/10.1002/adfm.201802317 and Yang et al. Bioinspired Slippery Lubricant-Infused Surfaces With External Stimuli Responsive Wettability: A Mini Review, https://doi.org/10.3389/fchem.2019.00826). It is recommended to add a statement to clearly separate the current work form these similar references and also define the review period (e.g. last five years).
Reviewer 3 Report
The paper “A review on smart lubricant-infused surface toward droplet manipulation” reviewed recent progress in smart lubricant-infused surface for droplet manipulation.
This paper shows a good review of lubricant-infused surface, there are some issues that need to address. But what’s the related subject of this manuscript with the scope of Nanomaterials?
what makes this review different from the others and from the most recent ones?
E.g.:
“Villegas, M., Zhang, Y., Abu Jarad, N., Soleymani, L., & Didar, T. F. (2019). Liquid-infused surfaces: a review of theory, design, and applications. ACS nano, 13(8), 8517-8536.
or
Li, J., Ueda, E., Paulssen, D., & Levkin, P. A. (2019). Slippery lubricant‐infused surfaces: properties and emerging applications. Advanced Functional Materials, 29(4), 1802317.“
More investigation or discussion on many dates from references, because the data may come from different assumptions or situation.
The future plan or future LIS development direction needs to be addressed. Section of drawbacks and future could be increased quality of the manuscript.
There are many grammatical errors, please carefully check the whole manuscript.
This review manuscript needs to at least two comparative tables with details.
Round 2
Reviewer 1 Report
The authors have implemented the changes related to the reviewers comments. The paper is now much easier to read, even if some more effort can be done to improve it further, concerning readability.
Reviewer 3 Report
Title:
A review on smart lubricant-infused surface toward droplet manipulationIn this revised manuscript, the Authors have made corrections according to referee comments. In my opinion, the manuscript in current form could be considered for acceptance.
